# Position: Profiling Game Worlds by Transition Complexity

**Lele Cao** [1]

## Abstract

Game world modeling (GWM) and reinforcement learning (RL) are often confounded because research papers rarely quantify how difficult the underlying transition prediction problem is at the declared interface (pixels/tokens/latents with finite history). We propose the Transition Complexity Profile (TCP): a small, reproducible set of metrics that characterizes an environment's (or gameplay dataset's) induced transition kernel by (i) intrinsic one-step branching, (ii) interaction-induced uncertainty and opponent influence when observable, and (iii) temporal/spatial dependency span via standardized probe curves. TCP is reported with an explicit reference distribution, protocol stochasticity, and a versioned measurement budget (sampling/resampling and fixed probe compute), enabling comparable numbers across benchmarks. We outline how common game families and modern "neural game engine" domains populate this landscape and call for TCP to become standard benchmark metadata and a required statistic in GWM and RL papers.

## 1. Introduction

Game world modeling (GWM) is quickly shifting from "model-based RL components" to interactive generative simulators. Early work such as (Ha & Schmidhuber, 2018) and latent-dynamics agents such as PlaNet/Dreamer (Hafner et al., 2019; 2021) established learned transition models as a practical substrate for control from pixels. Recent "neural game engine" efforts push this toward real-time, interactive rollouts: WHAM/WHAMM (Kanervisto et al., 2025) models gameplay and controller actions and demonstrate interactive transfer to Quake II (Microsoft Research, 2025), while DeepMind's Genie line targets promptable, real-time interactive worlds (Bruce et al., 2024). Relatedly, code-based world models translate interaction traces and rules

into executable simulators, offering a different axis of verifiability and control (Lehrach et al., 2025).

In parallel, discrete-token and diffusion world-model lines for Atari and beyond, such as IRIS (Micheli et al., 2023), DIAMOND (Alonso et al., 2024), further show that stronger generative predictors can support imagination and long-horizon rollouts. Despite this momentum, evaluation practice rarely reports how difficult the underlying transition prediction problem is at the declared interface. Instead, "world models" are primarily evaluated via:

- downstream return (when used for control),
- visual quality or qualitative rollouts (for generative models), and/or
- one-step prediction losses reported in incompatible units under incompatible tokenizations.

While useful, they do not answer a basic scientific question:

> ***How complex is the environment's transition kernel that we are asking the model to learn?***

Without a shared, operational notion of transition complexity, comparisons are confounded:

- A model that wins on Atari 100k may be solving a transition task whose branching, dependency span, and interaction uncertainty are fundamentally different from Quake II, Bleeding Edge, StarCraft, Minecraft, or prompt-generated interactive worlds.
- Even within Atari, the effective transition kernel depends on the evaluation protocol; for example, different ALE (Arcade Learning Environment) configurations introduce action-repeat ("sticky action") stochasticity with varying probabilities, yielding quantitatively different transition branching despite identical game logic.

Most importantly, benchmark[1] selection itself becomes ad hoc: without a shared "map" of environments by transition complexity, it is unclear which aspects of dynamics (branching, interaction, long-range dependence) different benchmarks actually stress, and whether current evaluations target the capabilities required by the next generation of GWM. Beyond evaluation, transition complexity can also inform ar-

---

[1]King AI Labs, Microsoft Gaming  Correspondence to: Lele Cao <lelecao@microsoft.com, lele.cao@king.com>.

*Proceedings of the $43^{rd}$ International Conference on Machine Learning*, Seoul, South Korea. PMLR 306, 2026. Copyright 2026 by the author(s).

---

[1]*benchmark* is used in a broad way to refer to a game, a learned GWM, or a dataset collected from gameplay, depending on context.

chitectural choice, as different world-model families handle branching and long-range dependencies differently.

This position paper argues that **GWM and RL research should report a standardized Transition Complexity Profile (TCP) for each game environment/dataset, quantifying transition branching, opponent-/human-induced uncertainty, and dependency span at the declared interface, alongside return, qualitative rollouts, and one-step prediction losses.**

**Scope of the position.** TCP is not proposed as a universal scalar notion of "game complexity". In particular, it does not measure Kolmogorov/rule-description complexity, computational complexity, solver difficulty, or the architecture-specific approximation error of a deterministic but highly nonlinear map. A low-entropy chaotic simulator may still be difficult for a particular predictor, and a high-entropy but simple sampler may be easy distributionally. The claim is narrower: TCP reports otherwise hidden properties of the induced next-interface prediction problem (branching, interaction uncertainty, and finite-history/spatial span) under a declared interface, reference distribution, and protocol.

## 2. Game Complexity and Related Work

This paper does not propose a new world-model architecture. Instead, it argues that current GWM and reinforcement learning (RL) research is missing a dedicated *measurement layer*: a way to characterize how difficult an environment's state transitions are to model, independent of a particular algorithm or reward objective.

### 2.1. Task and environment complexity in RL

There is substantial literature on "task" or "environment" complexity in RL, but it generally addresses different questions than the problem faced by a world model: predicting how the environment evolves from one state to the next.

- *Sample-complexity and regret measures* ask how difficult it is to *learn a good policy*, for example via quantities such as Eluder dimension (Russo & Van Roy, 2013), Bellman rank (Jiang et al., 2017), witness-rank-style parameters in contextual decision processes (Sun et al., 2019), or structural assumptions such as low-rank transition models (Agarwal et al., 2020). These notions govern exploration and control difficulty under a hypothesis class, but they do not directly quantify the *induced next-observation/next-state multiplicity* (branching) or the *finite-history dependence* faced by a world model at a declared interface.

- *State abstraction and bisimulation* study when different state representations preserve values or transition behavior (Li et al., 2006; Ferns et al., 2004); they focus on equivalence and compression, rather than providing a

small set of quantitative descriptors summarizing how complex the transition dynamics of an environment are.

- *Information-theoretic measures of process complexity*, such as predictive information or excess entropy (Bialek et al., 2001; Crutchfield & Feldman, 2003), quantify structure in time series. However, they are typically defined for passive stochastic processes, without actions, opponents, or standardized evaluation procedures, making them difficult to apply directly to interactive game environments and world-model benchmarks.

### 2.2. Classic game difficulty measures

Classic game analysis often characterizes difficulty via state-space size, game-tree branching factors (Shannon, 1950), and information-set structure (Kuhn, 1953), while computational game theory studies formal complexity classes for families of games (Hearn & Demaine, 2005). More recently, Bonnet et al. 2017 analyze the parameterized complexity of positional games, offering a complementary combinatorial view of game difficulty. In parallel, the Procedural Content Generation (PCG) community has begun standardizing benchmarks for generative challenges in games (Khalifa et al., 2025), focusing on content artifacts. Game-playing and planning work on state abstraction (Xu et al., 2022) further highlights that changing the representation alters the effective dynamics seen by a learner.

While valuable, these perspectives are not designed to yield a world-model-relevant characterization of the transition mapping $(S_t, A_t) \mapsto S_{t+1}$ (or its observation-level counterpart). A world model is fundamentally a transition model: without explicitly quantifying the structure and difficulty of the transition problem itself, progress across games, datasets, and modeling approaches remains difficult to interpret.

### 2.3. Evaluation practice in modern world models

Modern world-model papers typically report (i) downstream return when the model is used for control/planning, and/or (ii) predictive losses or qualitative rollouts as generative evidence (Hafner et al., 2019; 2021; Ha & Schmidhuber, 2018), consistent with broader model-based RL practice and taxonomy (Moerland et al., 2023). Recent "neural game engine" lines similarly emphasize interactive rollouts and visual fidelity, often under task- and protocol-specific choices (e.g., action stochasticity in ALE; tokenization and context windows in gameplay models Machado et al. 2018; Kanervisto et al. 2025; Bruce et al. 2024; Micheli et al. 2023; Alonso et al. 2024). TCP is not proposed as a replacement for these metrics; it is a *measurement layer* intended to make cross-domain and cross-protocol comparisons interpretable by reporting (a) intrinsic branching, (b) interaction-induced branching when observable, and (c) dependency span under a declared interface and measurement budget. This is

motivated by the long-standing view that learned dynamics quality (and its mismatch/uncertainty) can dominate planning and downstream behavior even when return-based evaluation is similar (Nagabandi et al., 2018).

## 2.4. A fixed seed does not trivialize the issue

One can make almost any simulator deterministic by conditioning on a random-number generator (RNG) seed or fixed opponent policy. However, this does not eliminate the scientific question of interest: what transition kernel is induced on the information actually available to the model or agent? World models almost never observe the full simulator state, RNG seed, or opponents' internal policies. They condition on an information interface (pixels, tokens, latents, truncated history), which induces an *effective* stochastic transition kernel driven by unobserved randomness and other agents' actions. This dependence on the observation interface is not a limitation: it reflects the true modeling problem and is precisely why transition complexity must be defined relative to the state representation any GWM operates on.

## 2.5. Relation to interpretability and causal views.

TCP is also aligned with interpretability goals: by decomposing uncertainty sources (chance vs. interaction) and quantifying where finite-history dependence matters, TCP helps localize why a world model is uncertain or brittle under a given interface and protocol. This complements causal and explainable RL perspectives that seek mechanistic, counterfactual accounts of agent behavior (Madumal et al., 2020).

## 3. Preliminaries

We want definitions broad enough to cover: board games (e.g., chess/go/tic-tac-toe), stochastic single-player games (e.g., match-3), imperfect-information games (e.g., card games), video games (e.g., Atari, Quake II), learned "neural environments" (e.g., WHAM-style playable models, Genie-style interactive world generators).

Let the underlying environment be a Markov game (stochastic game) with latent state $S_t \in \mathcal{S}$, our action $A_t \in \mathcal{A}$, opponent joint action $B_t \in \mathcal{B}$ (possibly depending on $A_t$ in sequential move settings), and exogenous "chance" variable $\Omega_t \in \mathcal{W}$. Assume a measurable function $F$ such that

$$S_{t+1} = F(S_t, A_t, B_t, \Omega_t),$$

where $\Omega_t$ captures all sources of randomness (including simulator RNG, procedural generation, random shuffles, etc.). For the discrete-time game settings considered here, any stochastic transition kernel can equivalently be represented as a deterministic function of the current state, actions, and an auxiliary random variable.

We distinguish between the latent environment state $S_t$,

which is internal to the game or simulator, and the observation $O_t$ available to the model at time $t$. A world model typically conditions on an information state $X_t$, constructed from the interaction history:

$$X_t = \phi(H_t), \quad H_t = (O_{1:t}, A_{1:t-1}, B_{1:t-1}),$$

where $\phi$ is a fixed state-construction mapping, and $B_{1:t-1}$ should be omitted in single-agent settings. In fully observed board games such as chess or Go, the information state $X_t$ can coincide with the full board configuration and is Markov. In Atari-like settings, $X_t$ is typically constructed by stacking the last $k$ observation frames to approximate a Markov state under partial observability. In high-dimensional gameplay modeling, such as WHAM, $X_t$ corresponds to a finite context window of past observation-action pairs (e.g., image or token sequences), which the model conditions on to predict the next observation.

This perspective is closely related to game state-abstraction work (Xu et al., 2022): changing $\phi$ alters the effective transition kernel induced at the model's observation interface, and TCP is intended to quantify that change operationally. The object a world model learns depends on the sufficiency of the chosen information state. In the fully observed Markov case, it targets

$$P(X_{t+1} \mid X_t, A_t),$$

where $X_{t+1}$ is obtained by applying the same mapping $\phi$ to the next interaction history. More generally, when $X_t = \phi(H_t)$ is not Markov, the true conditional is $P(X_{t+1} \mid H_t, A_t)$, and a finite-window world model targets an *operational Markov approximation*

$$P(X_{t+1} \mid X_t^{(k)}, A_t), \quad X_t^{(k)} = \phi_k(H_t),$$

where $\phi_k$ retains only a $k$-step truncated history (e.g., the last $k$ frames and actions). The dependency-span measurement introduced below tests how predictive performance changes as the retained history length $k$ increases.

This formulation is deliberately compatible with **partially observed settings**. Hidden simulator variables, beliefs, shuffled decks, or unobserved opponents remain in $S_t$, while the model only receives the finite information state $X_t^{(k)}$. At a fixed finite history, unresolved hidden variables appear as residual uncertainty in $P(X_{t+1} \mid X_t^{(k)}, A_t)$; increasing $k$ tests whether prediction improves as more history is supplied. If predictive performance continues to improve over all measured context lengths, the TCP report should state "no saturation within the evaluated range" rather than silently treating $X_t^{(k)}$ as Markov. Combinatorial interactions matter to TCP when they alter next-interface multiplicity or the length/structure of history needed for prediction. Purely long-horizon strategic difficulty, however, is complementary to TCP and should still be evaluated by return, planning, or game-theoretic metrics.

# 4. Transition Complexity Profile (TCP)

A single scalar "complexity" number can be misleading. We advocate a profile of complementary metrics, each mathematically well-defined, and each computable from logged transitions or instrumented simulators. Throughout, logarithms are base 2 (bits). For brevity, we use $X$ and $X'$ to denote $X_t$ and $X_{t+1}$ respectively.

## 4.1. Axis-I: Branching of the transition kernel

**Definition 1** (distributional one-step transition entropy). Fix a reference distribution $d(X, A)$ over encountered state-action pairs (e.g., the empirical dataset distribution or a standardized exploratory policy). Define

$$C_{\mathrm{H}}^{(1)}(d) = \mathbb{E}_{(X,A)\sim d}\big[H(X' \mid X, A)\big].$$

where $H(\cdot \mid \cdot)$ is conditional entropy.

**Definition 2** (effective one-step branching via collision entropy). For discrete $X'$, define the conditional Rényi entropy of order 2 (collision entropy)

$$H_2(X' \mid X{=}x, A{=}a) {=} -\log\sum_{x'} P(X'{=}x' \mid X{=}x, A{=}a)^2.$$

Define the effective one-step branching factor as the geometric mean (perplexity-like)

$$C_{\mathrm{B}}^{(1)}(d) = 2^{\mathbb{E}_{(X,A)\sim d}\big[H_2(X'\mid X,A)\big]}.$$

$2^{H_2}$ is the inverse collision probability and behaves like an effective support size of the next-state distribution, capturing the intrinsic multiplicity of possible next states under a given state-action pair. Using the geometric mean yields a perplexity-like statistic that is less sensitive to heavy tails.

Why this matters for GWM: modern world models already optimize log-loss or likelihood objectives, but they rarely report the intrinsic branching or multiplicity of the environment's transition dynamics as a property of the domain itself. A standardized $C_{\mathrm{H}}^{(1)} / C_{\mathrm{B}}^{(1)}$ therefore turns "how hard is the transition problem?" into a comparable statistic, analogous to perplexity in language modeling.

## 4.2. Axis-II: Interaction-induced branching

In many games, the dominant source of next-state uncertainty is not RNG; it is the behavior from other agents (i.e., stochasticity from bot/human opponents), including settings where competition induces emergent complexity in the interaction dynamics (Bansal et al., 2018).

**Definition 3** (opponent uncertainty under a population: within-policy vs. population heterogeneity). Let $\Pi$ be a distribution over opponent types $\theta$, with opponent policy

$\pi_B(\cdot \mid X, A; \theta)$ (allowing dependence on our action in sequential settings). Here $B$ denotes the opponent (joint) action at the same step as $(X, A)$. We consider the joint induced by $(X, A) \sim d$, $\theta \sim \Pi$, and $B \sim \pi_B(\cdot \mid X, A; \theta)$.

Define the *within-opponent* action entropy

$$C_{\mathrm{opp}}^{\mathrm{within}}(d, \Pi) = \mathbb{E}_{(X,A)\sim d}\Big[\mathbb{E}_{\theta\sim\Pi}\big[H(B \mid X, A, \theta)\big]\Big].$$

Define the *mixture* opponent action entropy (population-level uncertainty)

$$C_{\mathrm{opp}}^{\mathrm{mix}}(d, \Pi) = \mathbb{E}_{(X,A)\sim d}\big[H(B \mid X, A)\big],$$
$$P(B \mid X, A) = \mathbb{E}_{\theta\sim\Pi}\big[\pi_B(B \mid X, A; \theta)\big].$$

These quantities satisfy

$$H(B \mid X, A) = \mathbb{E}_{\theta}\big[H(B \mid X, A, \theta)\big] + I(\theta; B \mid X, A),$$

so $C_{\mathrm{opp}}^{\mathrm{mix}}$ captures both intrinsic stochasticity of each opponent and population heterogeneity, quantified by the conditional mutual information $I(\theta; B \mid X, A)$.

**Definition 4** (opponent influence on next state). Opponent entropy alone is not sufficient: if opponent actions have little effect on how the state evolves, interaction uncertainty is largely irrelevant for transition modeling. We define

$$C_{\mathrm{infl}}^{\mathrm{opp}}(d, \Pi) = \mathbb{E}_{(X,A)\sim d}\Big[\mathbb{E}_{\theta\sim\Pi}\big[I(B; X' \mid X, A, \theta)\big]\Big],$$

where $I(\cdot; \cdot \mid \cdot)$ denotes conditional mutual information under the induced joint distribution.

Together, $C_{\mathrm{opp}}^{\mathrm{within}}$ captures stochasticity within an individual opponent's policy (e.g., randomized play), $C_{\mathrm{opp}}^{\mathrm{mix}}$ captures uncertainty arising from facing a heterogeneous population of opponents, and $C_{\mathrm{infl}}^{\mathrm{opp}}$ quantifies how much this opponent variation actually induces branching in the next-state distribution. This separation is important: in chess, each move directly determines the next board state, yet the corresponding opponent influence on state transitions is huge.

In practice, multi-agent datasets often lack clean opponent types $\theta$ or stationarity. We recommend *tiered reporting*:

- $B_t$ observed and opponent type/ID $\theta$ available $\Rightarrow$ report $C_{\mathrm{opp}}^{\mathrm{within}}$, $C_{\mathrm{opp}}^{\mathrm{mix}}$, and $C_{\mathrm{infl}}^{\mathrm{opp}}$.
- $B_t$ observed but no reliable $\theta$ $\Rightarrow$ report only the mixture entropy $C_{\mathrm{opp}}^{\mathrm{mix}}$ (treating the dataset as the population) and the influence proxy $\widehat{C}_{\mathrm{infl}}^{\mathrm{opp}}$ from Sec. 6.4.
- $B_t$ not observed at the declared interface $\Rightarrow$ mark Axis-II as unobservable (do not infer $B_t$ from pixels).

This keeps Axis-II from becoming a source of brittle or non-reproducible inference in human gameplay logs. Moreover, $\Pi$ should be defined by a reproducible rule such as: (i) uniform over built-in bot difficulty settings; (ii) the empirical distribution over opponent IDs present in the dataset; or (iii)

a clustering of opponents into types with the clustering method and number of clusters reported.

Axis-II also interfaces naturally with recent work that *co-learns* strategic interaction structure (empirical games) alongside learned dynamics models, highlighting that opponent variability can be a first-class driver of the effective transition kernel (Smith & Wellman, 2024).

### 4.3. Axis-III: Dependency span

Even if $H(X'|X, A)$ is small, predicting $X'$ can require modeling long-range dependencies due to spatial coupling, cascades, occluded state, etc. We propose two operational dependency-span notions: one temporal, one spatial. Benchmarks can report one or both depending on domain.

#### 4.3.1. TEMPORAL DEPENDENCY SPAN

**Definition 5** (operational memory depth). Let $X_t^{(k)}$ denote a $k$-step truncated information state (e.g., last $k$ frames/actions). Define the best achievable predictive cross-entropy under a standardized probe model class $\mathcal{M}$:

$$\mathcal{L}_k = \inf_{q \in \mathcal{M}} \mathbb{E}\big[-\log q(X' \mid X_t^{(k)}, A_t)\big].$$

When the interface is tokenized, report $\mathcal{L}_k$ as bits per predicted token (the average negative log-likelihood per token); otherwise, $\mathcal{L}_k$ is reported per predicted next-observation unit under a declared discretization. Define the $\varepsilon$-memory depth (for a declared maximum context $K_{\max}$):

$$C_{\mathrm{mem}}(\varepsilon) = \min\left\{k \in \{1, \ldots, K_{\max}\} : \mathcal{L}_k \leq \min_{1 \leq j \leq K_{\max}} \mathcal{L}_j + \varepsilon\right\}.$$

To make $C_{\mathrm{mem}}(\varepsilon)$ comparable across domains, the probe model class $\mathcal{M}$ and training protocol must be fixed, analogous to standard linear probes in representation learning. Accordingly, TCP reports should evaluate $C_{\mathrm{mem}}(\varepsilon)$ using a small, versioned set of lightweight sequence predictors with fixed tokenization/interface, optimization settings, and compute budgets. The goal is not to recover an environment-invariant quantity, but to obtain a reproducible operational measure tied to a declared interface and measurement budget. To avoid arbitrary thresholding, we recommend reporting (i) the full curve $k \mapsto \mathcal{L}_k$ for $k \in \{1, 2, 4, 8, \ldots, K_{\max}\}$ and (ii) $C_{\mathrm{mem}}(\varepsilon)$ at two fixed thresholds, $\varepsilon \in \{0.01, 0.1\}$ bits/token. If other thresholds are used, these two values should also be reported for comparability.

**TCP-Ref probes (v1).** As a concrete standard, we define a versioned reference kit comprising at least two sequence probes trained on the declared interface: (i) TCP-Ref-GRU-v1, a 2-layer GRU (hidden size 512; token embedding 256; action embedding 64; dropout 0.1), and (ii) TCP-Ref-TX-v1, a decoder-only Transformer with 4 layers ($d_{\mathrm{model}}$=512, 8 heads, MLP width 2048, dropout 0.1) and learned positional

embeddings up to $K_{\max}$. Both use a linear decoder to the next-token vocabulary.

**Training protocol (v1).** All probes are trained with AdamW (lr $3 \times 10^{-4}$, $\beta$=$(0.9, 0.95)$, weight decay 0.1), gradient clipping at 1.0, and early stopping on held-out log-loss. TCP reports must include the probe version and training budget. These probes serve solely as measurement instruments and should not be replaced when architectures improve; new probe variants must be introduced as new versions.

This procedure is directly actionable: train the same standardized probes across increasing context lengths and report the smallest $k$ at which predictive performance saturates, as is common practice in high-dimensional GWMs.

#### 4.3.2. SPATIAL DEPENDENCY SPAN

For states with an explicit spatial or relational structure (e.g., boards, tiles, image patches, or spatially indexed tokens), how far influences propagate matters.

Let $X = (X^1, \ldots, X^n)$ denote state variables indexed by positions or regions (cells, tiles, image patches, spatial tokens, etc.), and let $N_r(i)$ denote the radius-$r$ neighborhood of index $i$ under a natural adjacency or distance metric defined by the interface.

**Definition 6** (operational spatial dependency radius via masked-neighborhood probes). Assume the state $X$ admits such a spatial or relational indexing. For a position $i$ and radius $r$, let $X_{N_r(i)}$ denote the subset of variables within the radius-$r$ neighborhood of $i$.

Define the best achievable predictive cross-entropy for the local next-variable using only the radius-$r$ neighborhood:

$$\mathcal{L}_{r,i} = \inf_{q \in \mathcal{M}_{\mathrm{loc}}} \mathbb{E}\big[-\log q(X'^i \mid X_{N_r(i)}, A)\big],$$

where $\mathcal{M}_{\mathrm{loc}}$ is a fixed local probe family, such as a masked Transformer or CNN (convolutional neural network) operating on local neighborhoods. Similar to the temporal case, we define the $\varepsilon$-dependency radius at position $i$ (for a declared $R_{\max}$):

$$C_{\mathrm{rad}}(i; \varepsilon) = \min\left\{r \in \{0, \ldots, R_{\max}\} : \mathcal{L}_{r,i} \leq \min_{0 \leq j \leq R_{\max}} \mathcal{L}_{j,i} + \varepsilon\right\}.$$

Aggregating the per-position radius $C_{\mathrm{rad}}(i; \varepsilon)$ via $\max_i$ (worst-case) or $\mathbb{E}_i$ (average over positions) yields a reproducible measure of spatial dependency span, even when exact conditional-independence assumptions are unrealistic (e.g., in pixel-based observations). For spatial $\varepsilon$, report the curves $r \mapsto \mathcal{L}_{r,i}$ (for representative or aggregated $i$) and $C_{\mathrm{rad}}(\cdot; \varepsilon)$ for $\varepsilon \in \{0.01, 0.1\}$ bits/token. For $\mathcal{M}_{\mathrm{loc}}$, use a fixed masked local predictor: a 4-layer masked Transformer over radius-$r$ neighborhood tokens ($d_{\mathrm{model}}$=256, 4 heads, MLP width 1024, dropout 0.1), trained with the same

AdamW protocol as in the temporal case. This local probe standardization is versioned as **TCP-Ref-Loc-v1**.

### 4.4. Small worked example: tic-tac-toe

At the full-board interface, define one TCP step as our legal move followed by the opponent response. If our move is terminal, $H(X'|X,A)=H_2(X'|X,A)=0$. Otherwise, if $m$ opponent replies are legal and the opponent is uniform random, $P(X'|X,A)$ is uniform over $m$ next boards, so $H(X'|X,A)=H_2(X'|X,A)=\log_2 m$. Under the random-play reference distribution (Appendix A.5), $C_{\mathrm{H}}^{(1)} \approx 1.906$ bits and $C_{\mathrm{B}}^{(1)} \approx 3.75$; the median and 90th percentile per-$(x,a)$ entropies are 2 and 3 bits. Because distinct opponent moves induce distinct next boards, $C_{\mathrm{infl}}^{\mathrm{opp}} \approx 1.906$ bits; because the full board is Markov, $C_{\mathrm{mem}}(\varepsilon)=1$. The point is not that tic-tac-toe is difficult, but that once $X$, $d$, and the step protocol are declared, every TCP number has an unambiguous meaning.

## 5. Sanity properties

A useful TCP should behave predictably under *refinement of state/interface* and *decomposition of uncertainty sources* (chance vs. interaction), consistent with Sec. 4's emphasis that TCP is defined at a declared interface $X_t = \phi(H_t)$ and reference distribution $d(X,A)$.

**Theorem 1** (Refinement cannot increase entropy for a fixed prediction target). Let $Y'$ be a fixed prediction target, and let $U$ and $V$ be random variables with $V=f(U)$ (so $V$ is a deterministic coarsening of $U$). Then for any action $A$,

$$H(Y' \mid U, A) \leq H(Y' \mid V, A).$$

**Proof.** Since $V = f(U)$, conditioning on $U$ is at least as informative as conditioning on $V$. Equivalently, $(V, A)$ is a function of $(U, A)$, and conditional entropy is monotone under additional conditioning:

$$H(Y' \mid V, A) = H(Y' \mid f(U), A) \geq H(Y' \mid U, A). \quad \square$$

This theorem should be read as a same-target statement: refining the conditioning information cannot increase uncertainty about a fixed next-variable $Y'$. It does not, by itself, order TCP values computed at two different interfaces when the predicted next-interface variable also changes, e.g., predicting $U'$ at a refined interface versus predicting $V'$ at a coarsened interface.

TCP is a property of the triple *(environment, interface, reference distribution)*. For cross-paper comparisons, we recommend: (1) report TCP at the model's native interface $X$ (incl. the chosen history window), and (2) when a community-standard public interface exists (e.g., ALE frames for Atari; board state for chess/go), also report TCP at that public interface under the same declared $d(X,A)$. Direct numerical comparison of TCP values requires matching the interface, protocol, and reference distribution. When one interface is a deterministic coarsening of another, Theorem 1 only justifies same-target entropy comparisons, not a general scalar ordering between full TCP profiles at different interfaces.

**Theorem 2** (Unobserved randomness upper-bounds next-state uncertainty). Assume that the next interface state is induced by an underlying deterministic update with chance and (possibly) opponent actions:

$$X' = G(X, A, B, \Omega),$$

for a measurable function $G$, where $B$ denotes the opponent joint action at the step and $\Omega$ collects all exogenous randomness (RNG, shuffles, procedural draws, etc.). Then

$$H(X' \mid X, A) \leq H(B, \Omega \mid X, A).$$

Moreover, letting $Z = (B, \Omega)$,

$$\begin{aligned} H(X' \mid X, A) &= I(Z; X' \mid X, A), \\ &= H(Z \mid X, A) - H(Z \mid X, A, X'). \end{aligned}$$

**Proof.** Fix $(X, A)$ and let $Z = (B, \Omega)$. Since $X' = G(X, A, Z)$ is deterministic given $(X, A, Z)$, we have $H(X' \mid X, A, Z) = 0$. Therefore,

$$I(Z; X' \mid X, A) = H(X' \mid X, A) - H(X' \mid X, A, Z).$$

Since $H(X' \mid X, A, Z) = 0$, it follows that

$$I(Z; X' \mid X, A) = H(X' \mid X, A).$$

which gives the first equality. The second equality follows from the identity $I(Z; X' \mid X, A) = H(Z \mid X, A) - H(Z \mid X, A, X')$. $\square$

When logs expose chance variables and/or opponent actions, reporting source entropies provides interpretable diagnostics. In particular, reporting the joint source entropy $H(B, \Omega \mid X, A)$ yields an explicit upper bound on the intrinsic one-step uncertainty $H(X' \mid X, A)$ via Theorem 2; reporting $H(\Omega \mid X, A)$ and $H(B \mid X, A)$ separately further helps attribute uncertainty to chance vs. interaction. The residual $H(B, \Omega \mid X, A, X')$ quantifies irrelevant randomness: variability in $(B, \Omega)$ that does not change $X'$ at the declared interface, clarifying when high opponent/chance entropy does not translate into high transition branching.

## 6. Operational aspects of TCP

TCP is intended as a domain/benchmark statistic reported alongside environment and dataset details, not as an additional hidden tuning axis.

**The measurement budget.** Axes II–III can require simulator resampling and/or auxiliary probe training. To keep

TCP comparable and reproducible, every TCP report must declare its *measurement budget* (e.g., number of sampled $(x, a)$ pairs; simulator resamples $M$ per pair when applicable; probe architectures/version strings; training budget in tokens/updates; and the exact interface/tokenization and context window). This makes TCP an explicit instrumentation layer rather than an unreported burden. We recommend two versioned tiers:

- TCP-Lite-v1: sample 5,000 $(x, a) \sim d$; if reset + resampling is available use $M{=}8$ i.i.d. resamples per $(x, a)$; train each TCP-Ref probe for a fixed 200M tokens (or a fixed update budget stated in the report), using 3 seeds and reporting mean±std.
- TCP-Std-v1: sample 20,000 $(x, a) \sim d$; use $M{=}32$ resamples when available; train each TCP-Ref probe for 1B tokens (or a matched update budget), using 3 seeds and reporting mean±std plus 95% bootstrap CIs over $(x, a)$ pairs or trajectories.

Benchmarks should publish TCP-Std-v1 values once; model papers may report TCP-Lite-v1 under limited compute, but must declare the tier. Because TCP depends on the reference distribution $d(x, a)$, report TCP under (i) $d_{\text{data}}$ (the training/behavior distribution), and (ii) a standardized probing distribution $d_{\text{probe}}$ (e.g., scripted exploration or random/no-op mixtures where appropriate). In addition to means, report median and 90th percentile of per-$(x, a)$ quantities to avoid masking rare high-branching regimes.

The fixed sample counts are comparable measurement budgets, not variance guarantees. In high-branching regimes, a tier may yield collision-limited estimates or wide confidence intervals; this should be reported as part of the TCP result rather than hidden by increasing the budget ad hoc. Authors may additionally report higher-budget estimates, but the versioned TCP-Lite-v1/TCP-Std-v1 values should remain visible for comparability. The dependence on $d$ is also intentional: $d_{\text{data}}$ reports the transition problem actually seen by the model, while $d_{\text{probe}}$ provides a shared slice for cross-paper comparison. Direct numerical comparison requires matching the interface, protocol, and reference distribution.

### 6.1. Simulator access: reset + controllable randomness

If the simulator can reset to a specific latent state consistent with the declared interface and can vary exogenous randomness (and, in multi-agent settings, can sample opponents from $\Pi$), then for each sampled $(x, a) \sim d$:

1. Reset to the underlying state corresponding to $x$ (at the declared interface).

2. Apply action $a$ and generate $M$ i.i.d. next-interface samples $x'_1, \ldots, x'_M$ by resampling $\Omega$ (and, if applicable, sampling opponents from $\Pi$).

3. Estimate per-$(x, a)$ branching:

- Estimate the collision probability

$$\widehat{p}_{\text{coll}}(x, a) = \frac{1}{M(M-1)} \sum_{1 \leq i \neq j \leq M} \mathbf{1}\left[x'_i = x'_j\right],$$

which is unbiased for $\sum_{x'} P(x' \mid x, a)^2$ under i.i.d. resampling, and report the following as a primary estimator for Definition 2.

$$\widehat{H}_2(X' \mid x, a) = -\log \widehat{p}_{\text{coll}}(x, a).$$

If $\widehat{p}_{\text{coll}}(x, a){=}0$ (no collisions), report $\widehat{p}_{\text{coll}} = 0$ and the resolution threshold $H_{2,\text{res}}(M) = \log \binom{M}{2}$; mark the estimate as collision-limited, not a finite-sample lower bound.

- Optionally estimate $\widehat{H}(X' \mid x, a)$ (Definition 1) when $X'$ is discrete at the interface. For high-dimensional observations (e.g., pixels) or continuous latents, report collision-based branching $\widehat{H}_2$ (Definition 2) on a declared discretization of $X'$ (e.g., a fixed tokenizer) as the primary Axis-I estimator, and treat Shannon-entropy estimates as optional and interface-dependent.

For Axis II (when $B$ is observable/controllable), collect sampled opponent actions $b$ and compute $C_{\text{opp}}^{\text{within}}$, $C_{\text{opp}}^{\text{mix}}$ (Definition 3) and $C_{\text{infl}}^{\text{opp}}$ (Definition 4) under the same declared $(d, \Pi)$.

### 6.2. Log-only benchmarks: probe-based proxies

If only logged gameplay is available (typical for WHAM- or Genie-style training), one cannot resample $X'$ at a fixed $(x, a)$. TCP remains computable as operational quantities tied to a fixed probe family/budget:

- Axis I proxy (branching via predictive coding). Train standard probes and report cross-entropy $\mathcal{L} = \mathbb{E}[-\log q(X' \mid X, A)]$ in bits per token on the declared interface. This upper-bounds $H(X' \mid X, A)$ under $d_{\text{data}}$ (cross-entropy = entropy + KL), but may be probe-limited; therefore report at least two probe families (TCP-Ref-GRU-v1 and TCP-Ref-TX-v1) under the declared tier.
- Axis III (memory depth). Compute $\mathcal{L}_k$ across context lengths and report the curve $k \mapsto \mathcal{L}_k$ plus $C_{\text{mem}}(\varepsilon)$ at $\varepsilon \in \{0.01, 0.1\}$ bits/token, using the fixed TCP-Ref probes and budgets from Sec. 4.

Axis II should be reported only when $B_t$ is logged at the declared interface (do not infer opponent actions from pixels). For log-only TCP, include a coverage note: number of trajectories/transitions or tokens, behavior source/policy when known, train/validation/test split, and any filtering. The resulting TCP is a property of the logged dataset slice under $d_{\text{data}}$, not an environment-wide constant. For raw video or gameplay logs with no $B_t$, mark Axis II as unobservable; do not infer opponent actions from pixels. Multi-actor effects still contribute to Axis-I effective branching and Axis-III

memory requirements, but the source decomposition is not instrumented.

## 6.3. Protocol sensitivity: declare injected stochasticity

TCP is defined for the induced transition kernel at a declared interface. Any benchmark protocol that modifies action execution or injects randomness (e.g., action repeat/stickiness, frame-skip rules, randomized resets, no-op starts) changes that kernel and thus changes TCP. Consequently, TCP reports must specify the full transition protocol (incl. parameter values), otherwise two results on the "same" benchmark may correspond to different kernels and are not directly comparable. Atari/ALE is a common example: sticky-action probability and frame-skip conventions vary across implementations and materially affect one-step branching.

*Sanity check: sticky actions yield a one-step lower bound.* Under a sticky-action protocol where the executed action repeats the previous executed action with probability $p$ and equals the chosen $A_t$ with probability $1-p$, if the underlying simulator is deterministic given the executed action, then for any $(x, a)$ where the two candidate next-interface states differ and the interface omits the previous executed action, the induced distribution has two outcomes with probabilities $(1 - p)$ and $p$, hence $H(X' \mid X=x, A=a) = h_2(p)$. For $p=0.25$ as commonly used in ALE protocols (Machado et al., 2018), this contributes $h_2(0.25) \approx 0.811$ bits in those regimes, illustrating why protocol details are part of TCP.

## 6.4. Minimal TCP table: required fields

For each benchmark, report TCP under $d_{\text{data}}$ and $d_{\text{probe}}$, together with the declared interface $X_t$ (board/pixels/tokens), tokenizer (if any), and context window.

- Axis I: simulator case $\Rightarrow$ mean/median/90th of $\widehat{H}_2(X' \mid x, a)$ (and $\widehat{H}$ when well-defined at the interface); log-only case $\Rightarrow$ probe cross-entropy $\mathcal{L}$ (bits/token) for TCP-Ref-GRU-v1 and TCP-Ref-TX-v1.
- Axis II: if $(B, \theta)$ available $\Rightarrow$ $C_{\text{opp}}^{\text{within}}$, $C_{\text{opp}}^{\text{mix}}$, and $C_{\text{infl}}^{\text{opp}}$; if $B$ available but no reliable $\theta \Rightarrow$ report $C_{\text{opp}}^{\text{mix}}$ and an influence proxy (below); if $B$ unobserved $\Rightarrow$ mark Axis II "unobservable".
- Axis III: report $k \mapsto \mathcal{L}_k$ plus $C_{\text{mem}}(\varepsilon)$ at $\varepsilon \in \{0.01, 0.1\}$; when a spatial index is meaningful, report $C_{\text{rad}}(\cdot; \varepsilon)$ via TCP-Ref-Loc-v1 aggregated by $\max_i$ and $\mathbb{E}_i$.

Train two fixed-budget predictors $q_0(X' \mid X, A)$ and $q_1(X' \mid X, A, B)$ under the same probe family and report

$$\widehat{C}_{\text{infl}}^{\text{opp}} = \mathbb{E}[-\log q_0(X' \mid X, A)] - \mathbb{E}[-\log q_1(X' \mid X, A, B)].$$

With well-specified probes this estimates $\mathbb{E}[I(B; X' \mid X, A)]$ (Definition 4); in all cases it is a reproducible operational influence measure tied to the declared interface and probe budget.

## 6.5. Reliability requirements (TCP-Ref v1.0)

TCP reports must include: (i) probe version strings (TCP-Ref-GRU-v1/TCP-Ref-TX-v1/TCP-Ref-Loc-v1), (ii) full measurement budget (sampled $(x, a)$, $M$, tokens/updates, seeds), (iii) interface/tokenization details (including determinism), and (iv) 3-seed mean±std for probe-based quantities. Report 95% bootstrap CIs over $(x, a)$ pairs (simulator) or trajectories (logs) for each mean statistic. When resampling is limited and $X'$ is large, prefer collision-based branching ($H_2$) as primary; estimate collision probability with the unbiased pairwise-collision estimator above and bootstrap for uncertainty. Finally, include a single probe-capacity sanity check: widen TCP-Ref-TX by $2\times$ at the same budget and report the loss change; if improvement exceeds 0.1 bits/token, flag the corresponding proxy as *probe-limited at this tier* and interpret it as an upper bound.

# 7. Call to Action: TCP as Benchmark Meta

TCP becomes useful only if it is reported as benchmark metadata rather than as an optional post-hoc analysis. We therefore propose the following division of labor.

**Benchmark maintainers.** For each public benchmark-protocol-interface triple, publish a *TCP card*: a kernel declaration (environment/version, declared interface $X_t$, reference distributions, and transition protocol), canonical $d_{\text{probe}}$, TCP-Std-v1 values with uncertainty, the tokenization/preprocessing used to define $X_t$, and scripts sufficient to reproduce the sampling and probe evaluation. Simulator-backed benchmarks should expose reset/replay hooks, controllable resampling of exogenous randomness when possible, executed-action logs, and opponent-action/type fields when available. If these fields are not exposed, the TCP card should say so explicitly rather than leaving the missing source of uncertainty implicit.

**Model and dataset papers.** Every GWM/RL paper should include the minimal TCP table from Sec. 6.4 for each environment or gameplay dataset. If a paper uses an unmodified public setup with a published TCP-Std-v1 card, it may cite that card for $d_{\text{probe}}$ and only report TCP-Lite-v1 under $d_{\text{data}}$ when the behavior distribution differs. If the paper changes the induced kernel – for example by changing tokenization, context window, sticky-action probability, frame-skip, wrappers, reset distribution, or the modeled interface – it should recompute at least TCP-Lite-v1 for that modified setup.

**Suite designers.** Benchmark suites should be curated to span TCP axes, not only modalities or reward tasks. A useful suite should include Axis-I-dominant uncertainty, Axis-II-dominant interaction uncertainty when observable, mixed chance-and-interaction settings, and high temporal/spatial dependency span. Paired variants such as sticky actions on/off, deterministic vs. stochastic resets, or board-state vs.

pixel interfaces should be treated as distinct induced kernels and reported separately.

**Worked examples as regression tests.** The tic-tac-toe calculation in Sec. 4.4 and Appendix A.5, together with the sticky-action calculation in Sec. 6.3, are intended as small regression tests for the protocol. Tic-tac-toe shows that once $X$, $d$, and the step definition are fixed, Axis-I and Axis-II values are exactly interpretable. Sticky actions show that changing a wrapper while keeping the game name fixed changes the induced transition kernel; for $p = 0.25$, the one-step contribution is $h_2(0.25) \approx 0.811$ bits whenever the two executed-action outcomes differ.

**Reference implementation plan.** The practical artifact needed for adoption is a lightweight TCP-Ref implementation, not a new modeling framework. The planned reference package should contain: (i) collision-entropy and cross-entropy estimators with bootstrap confidence intervals; (ii) TCP-Ref-GRU-v1, TCP-Ref-TX-v1, and TCP-Ref-Loc-v1 configuration files and training scripts; (iii) a JSON/YAML TCP-card schema plus a LaTeX table generator for Sec. 6.4; and (iv) unit-test examples for tic-tac-toe and sticky-action protocols. A companion benchmark-card repository should collect versioned TCP-Std-v1 values contributed by benchmark maintainers. This separates one-time canonical measurement from per-paper recomputation and makes adoption feasible for smaller labs.

## 8. Alternative Views

**"Return is the only metric that matters."** If a world model yields high downstream return when used for control or planning, its fidelity to the true transition kernel is irrelevant.

*Response.* Return is necessary but not sufficient. As shown by standard identifiability arguments, distinct transition kernels can induce identical returns under a fixed policy and evaluation protocol. Moreover, world models are increasingly reused beyond a single policy – for transfer, editing, counterfactual reasoning, or as interactive simulators. In these settings, miscalibrated transition structure (e.g., incorrect branching or dependence length) can dominate performance despite comparable returns.

**"Transition stochasticity is an artifact of representation".** Any stochastic simulator can be made deterministic by augmenting the state with a random seed or hidden variables; therefore, transition stochasticity is not intrinsic.

*Response.* While true at the simulator level, this misses the relevant object for learning systems: the effective transition kernel induced on the agent's information state $X_t$. If the seed or latent variables are unobserved (as in nearly all practical settings), the induced kernel on $X_t$ remains stochastic. TCP is explicitly defined relative to the state interface used

by the model, making representation dependence a feature rather than a flaw.

**"Structured transition models already address this".** Factored MDPs, dynamic Bayesian networks, and related structured-dynamics models already capture transition structure; no new complexity framework is needed.

*Response.* These frameworks exploit conditional independence and locality when they exist. The environments emphasized in this paper (resolver-based, event-driven games with cascades and long-range effects) provide natural counterexamples where bounded-locality assumptions fail. TCP does not replace structured modeling; it provides a way to *diagnose* when such assumptions are likely to succeed or fail, and to design benchmarks that expose these regimes.

**"Entropy is not all forms of complexity."** Entropy can be small for deterministic but nonlinear or chaotic dynamics; it treats distinct interface states as distinct even when they are semantically equivalent; and it does not by itself measure planning or solver difficulty.

*Response.* TCP profiles the induced transition-prediction problem at a declared interface and reference distribution. It measures branching, interaction effects, and dependency span; it does not measure rule-description/Kolmogorov complexity, computational complexity, solver difficulty, or model-class approximation error. Semantic equivalence must be handled by the declared interface or tokenizer, and strategic difficulty should still be reported through return, planning, or game-theoretic metrics.

## 9. Conclusion

Games, world models, and reinforcement learning increasingly revolve around the same object: the transition kernel at a declared interface (pixels/tokens/latents with finite history). Yet today we rarely report how hard that transition prediction problem actually is, so comparisons across benchmarks, protocols, and representations are easily confounded.

We argued for a simple fix: make transition complexity a standard, reproducible piece of benchmark metadata. The proposed Transition Complexity Profile (TCP) summarizes (I) one-step branching, (II) interaction-induced uncertainty and opponent influence when observable, and (III) temporal/spatial dependency span via standardized probe curves, all under an explicit interface, reference distribution, protocol, and versioned measurement budget.

TCP is not a replacement for return or qualitative rollouts; it is the missing measurement layer that makes them interpretable. If adopted, it would enable principled cross-domain comparisons, more deliberate benchmark design, and clearer claims about what modern "neural game engines" and world models have truly learned.

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

## A. Mapping common game domains into the TCP landscape

This appendix gives a qualitative map of common game families in the Transition Complexity Profile (TCP) space (Sec. 4). The entries are schematic qualitative bins, intended only to illustrate how different domains may emphasize different TCP axes. Quantitative benchmark releases should instead report measured TCP values under a declared interface $X_t$, reference distribution $d(x, a)$ (at least $d_{\text{data}}$ and $d_{\text{probe}}$), protocol, and measurement tier.

### A.1. Axes and reporting convention

We summarize the TCP axes using the paper's notation:

- **Axis I (one-step branching)**: $C_{\text{H}}^{(1)}(d)$ and/or $C_{\text{B}}^{(1)}(d)$ (simulator), or probe cross-entropy proxies (log-only).

- **Axis II (interaction)**: report $C_{\text{opp}}^{\text{mix}}(d, \Pi)$ (and $C_{\text{opp}}^{\text{within}}$ when types are available) plus $C_{\text{infl}}^{\text{opp}}(d, \Pi)$ *only if* opponent actions $B_t$ are observed at the declared interface; otherwise mark as **–** (unobservable), per Sec. 6.4.

- **Axis III (dependency span)**: $C_{\text{mem}}(\varepsilon)$ and, when a spatial index is meaningful, $C_{\text{rad}}(\cdot; \varepsilon)$ via the TCP-Ref probe curves.

Unless noted, entries assume the typical world-model interface in that domain: observation (board/pixels/tokens) plus *ego* action; other agents' controls are treated as latent unless explicitly logged.

### A.2. Landscape table

Table 1 summarizes how common game and world-model benchmark families qualitatively populate the Transition Complexity Profile (TCP) space along Axes I to III. The Low/Med/High entries are schematic qualitative bins intended to build intuition, not computed TCP measurements. The table is meant as a reading guide rather than a taxonomy: its purpose is to highlight where different domains concentrate transition difficulty (branching, interaction, or dependency span), and to expose gaps or redundancies in current benchmark coverage.

### A.3. Diagnostic implication

Table 1 highlights a common confound: many standard world-model benchmarks are effectively single-agent (Axis II unobservable by design) with moderate one-step branching and moderate dependency span, while emerging "neural game engine" directions concentrate in high-span regimes and may hide multi-actor uncertainty inside Axis I when other controls are not part of the interface. TCP turns these differences into explicit, reportable benchmark metadata.

### A.4. Using the map in practice

To make this landscape quantitative (and comparable across papers), include a small TCP block alongside benchmark specs:

1. Declared interface $X_t$ (incl. tokenization and context length) and full transition protocol.

2. Reference distribution(s) $d_{\text{data}}$ and $d_{\text{probe}}$ and the TCP tier/budget (TCP-Lite-v1 or TCP-Std-v1).

3. Axis I: $C_{\text{H}}^{(1)}(d)$ / $C_{\text{B}}^{(1)}(d)$ (simulator) or probe cross-entropy proxies (log-only).

4. Axis II (if observable): $C_{\text{opp}}^{\text{mix}}$ (and within when available) and $C_{\text{infl}}^{\text{opp}}$; otherwise mark "unobservable".

5. Axis III: the probe curves and $C_{\text{mem}}(\varepsilon)$ (and $C_{\text{rad}}$ when meaningful).

| Domain / family | Axis I | Axis II | Axis III | World-model relevance (what TCP typically diagnoses) |
|---|---|---|---|---|
| Tic-tac-toe | Med | Med | Low | Deterministic rules; uncertainty comes from opponent replies. Useful as a TCP instrumentation sanity check (Sec. A.5). |
| Chess / Go | High | High | Low–Med | No chance; effective branching is dominated by opponent population. Markov at full-board interface; "span" mainly reflects nonlocal move effects / representation choices. |
| Card games (poker-like) | High | High | High | Clean chance vs. strategic uncertainty; imperfect information makes history/belief dependence central at common interfaces. |
| Match-3 (Candy-Crush-like) | High | – | Med–High | Chance-driven branching from spawns; cascades/resolvers induce wide spatial coupling within a step. |
| Atari (ALE) | Low–Med | – | Med | Transition branching is protocol-dependent (sticky actions, frame-skip, reset rules); partial observability drives memory depth at pixel interfaces. |
| WHAM / Bleeding Edge-style gameplay logs | Med | – | High | Log-only regime: Axis I/III are typically probe-based at token interfaces; multi-actor effects often appear as latent uncertainty when other controls are not logged. |
| WHAMM / Quake II-style interactive modeling | Med | – | High | Real-time rollouts stress long-horizon calibration and $C_{\mathrm{mem}}$ saturation under fixed context windows. |
| Promptable interactive worlds (Genie-style) | Med–High | – | High | Open-ended generation emphasizes controllability + minutes-long coherence; TCP should be reported at the model's native token/latent interface. |
| Counter-Strike-like video/gameplay modeling | Med–High | Med–High | High | Multi-actor dynamics can drive large effective branching; whether Axis II is measurable depends on whether other agents' actions are exposed vs. latent. |

*Table 1.* Schematic TCP landscape map (illustrative only). "Low/Med/High" bins are qualitative, non-measured summaries and can shift with the declared interface, protocol, and $d(x, a)$; quantitative comparisons require measured TCP-Lite-v1/TCP-Std-v1 values (Sec. 6.4).

### A.5. Worked numeric TCP example: tic-tac-toe under a fixed probe policy

We fully compute TCP for tic-tac-toe under an explicit interface and $d(x, a)$.

**Interface.** Let $X_t$ be the full $3 \times 3$ board (fully observed Markov) and $A_t$ a legal move for the current player. We define one step as "our move plus the opponent response": given $(X_t, A_t)$, $X_{t+1}$ is the board after applying our move and then the opponent move (if nonterminal after our move).

**Reference distribution $d_{\mathrm{probe}}$.** Start from the empty board; both players act uniformly at random among legal moves until termination. Let $d_{\mathrm{probe}}(x, a)$ be the induced distribution over encountered $(x, a)$ on our turns.

**Opponent population $\Pi$.** A singleton uniform-random opponent (so $C_{\mathrm{opp}}^{\mathrm{within}} = C_{\mathrm{opp}}^{\mathrm{mix}}$).

**Axis I (one-step branching).** Under $d_{\mathrm{probe}}$,

$$C_{\mathrm{H}}^{(1)}(d_{\mathrm{probe}}) = \mathbb{E}[H(X' \mid X, A)] \approx 1.906 \text{ bits.}$$

Per-$(x, a)$ entropies lie in $\{0, 1, 2, \log_2 6, 3\}$, with median 2 bits and 90th percentile 3 bits. Because the conditional next-state distributions are uniform over opponent legal replies

when nonterminal, $H_2(X' \mid X, A) = H(X' \mid X, A)$, so

$$C_{\mathrm{B}}^{(1)}(d_{\mathrm{probe}}) = 2^{\mathbb{E}[H_2(X' \mid X, A)]} \approx 2^{1.906} \approx 3.75.$$

**Axis II (interaction + influence).** There is no chance at this interface; all branching comes from the opponent:

$$C_{\mathrm{opp}}^{\mathrm{within}}(d_{\mathrm{probe}}, \Pi) = C_{\mathrm{opp}}^{\mathrm{mix}}(d_{\mathrm{probe}}, \Pi) \approx 1.906 \text{ bits.}$$

Since $X'$ is a deterministic function of $(X, A, B)$ and distinct opponent moves induce distinct next boards,

$$C_{\mathrm{infl}}^{\mathrm{opp}}(d_{\mathrm{probe}}, \Pi) = \mathbb{E}[I(B; X' \mid X, A)] \approx 1.906 \text{ bits.}$$

**Axis III (dependency span).** With the full-board interface the process is Markov, so $C_{\mathrm{mem}}(\varepsilon) = 1$ for any $\varepsilon > 0$; the spatial radius is bounded by the $3 \times 3$ board.

**Takeaway.** Once $X_t$, $d(x, a)$, and the transition protocol are declared, TCP can be reported numerically and (here) cleanly attributes one-step branching to opponent-induced uncertainty.

# B. Research agenda and community recommendations

Sections 4 to 6.4 define TCP as a *measurement layer* for game dynamics: a reproducible characterization of the induced transition kernel at a declared interface $X_t = \phi(H_t)$, under a declared reference distribution $d(X, A)$, and under a declared protocol (wrappers, stochasticity injection, etc.). The remaining question is adoption: how do we make TCP a routine, low-friction part of world-model and benchmark practice? We propose three concrete community actions: (i) standardize TCP reporting in papers, (ii) curate suites that intentionally span TCP axes, and (iii) treat TCP-relevant instrumentation as benchmark metadata, not an optional extra.

## B.1. TCP in papers

**Recommendation (minimum).** Any paper that trains or evaluates a game world model (or uses a learned world model as an environment) should include a TCP report for each benchmark. Concretely, the report should consist of: (i) the *minimal TCP table* from Sec. 6.4 (Axis I–III, with uncertainty), and (ii) a short *kernel declaration* that makes the induced transition kernel unambiguous.

**Kernel declaration (required header).** Because TCP is a property of $(\text{environment}, \text{interface}, d)$ and is protocol-sensitive (Sec. 6), a TCP report is only interpretable if the following are explicitly stated:

- **Environment and version.** Benchmark name, exact version/commit, and any wrappers that modify transitions (e.g., sticky-action probability, frame-skip/action-repeat, randomized no-op starts, reset randomization).
- **Declared interface.** The modeled information state $X_t$ (pixels/board/tokens/latents), including any preprocessing and, crucially, the history window/context construction $\phi_k$ (Sec. 3).
- **Reference distributions.** The two distributions under which TCP is computed: $d_{\text{data}}$ (behavior/training distribution) and $d_{\text{probe}}$ (standardized probing distribution) as defined in Sec. 6.
- **TCP tier and budget.** The tier string (TCP-Lite-v1 or TCP-Std-v1), including the sampled $(x, a)$ count, resamples $M$ when applicable, probe version strings (TCP-Ref-GRU-v1/TCP-Ref-TX-v1/TCP-Ref-Loc-v1), and training budget (tokens/updates), exactly as required in Sec. 6.4.
- **Axis-II observability.** Whether $B_t$ (opponent action) and $\theta$ (opponent identity/type) are available *at the declared interface*; if not, Axis II must be marked unobservable (Sec. 4.2).

**Numbers (the TCP table).** Given the above header, report the Axis I/II/III quantities exactly as specified in Sec. 6.4,

including mean/median/90th percentiles where defined and uncertainty (3-seed mean±std for probe-based quantities; bootstrap CIs as specified). In particular:

- If simulator reset/resampling is available, Axis I should be reported using collision-based branching $\widehat{H}_2(X' \mid x, a)$ as primary (Sec. 6.1).
- If only logs are available, Axis I is reported as probe cross-entropy $\mathcal{L}$ (bits/token) under TCP-Ref probes (Sec. 6.2), clearly labeled as a proxy/upper bound.
- Axis III should include the full $k \mapsto \mathcal{L}_k$ curve and $C_{\text{mem}}(\varepsilon)$ at $\varepsilon \in \{0.01, 0.1\}$ (Sec. 4.3), using the fixed probe versions/budgets.

**Practical division of labor (cite vs. recompute).** To keep reporting low-friction, we recommend the following norm:

- If benchmark maintainers publish TCP-Std-v1 under a standardized $d_{\text{probe}}$ and protocol, model papers may *cite* those TCP-Std-v1 numbers directly for context, and only compute TCP-Lite-v1 under $d_{\text{data}}$ if needed.
- If a paper changes the protocol/interface (e.g., different stickiness, different tokenization, different context construction $\phi_k$), it has changed the induced kernel and must recompute TCP (at least TCP-Lite-v1) for that modified setup.

**Why this belongs in papers.** Downstream return and qualitative rollouts remain essential, but TCP supplies the missing denominator: it tells the reader *which transition problem* the model was asked to solve at the stated interface, under the stated protocol and distribution. This directly addresses the confounds highlighted in Sec. 1 (cross-domain comparisons and protocol sensitivity).

## B.2. TCP-spanning suites

**Recommendation (design principle).** Benchmark suites for GWMs should be curated to *span the TCP axes*, not merely to vary modality (pixels vs. tokens) or reward structure. The goal is a suite where different regions of the TCP landscape are intentionally represented, so architectural and objective choices can be evaluated against specific transition-kernel properties.

**A minimal TCP-spanning suite.** At minimum, a suite should include benchmarks (or benchmark *variants*) that realize distinct combinations of:

- **Axis I-dominant uncertainty:** high $\widehat{H}_2(X' \mid x, a)$ under low or absent interaction uncertainty (e.g., chance-heavy puzzles or spawn-driven dynamics).
- **Axis II-dominant uncertainty:** low chance-driven branching but high opponent influence $C_{\text{infl}}^{\text{opp}}(d, \Pi)$ under a declared $\Pi$ (e.g., strong dependence on opponent actions in competitive games).

- **Mixed chance + interaction:** both intrinsic branching and interaction-induced branching are substantial (e.g., stochastic imperfect-information games with strategic opponents).
- **Long dependency span:** large $C_{\mathrm{mem}}(\varepsilon)$ (and/or large spatial radii when meaningful), emphasizing partial observability, delayed consequences, and long-range coupling.

**Protocol and interface variants are part of the suite.** Because TCP is protocol-sensitive (Sec. 6.3 to 6.4), suites should include *paired variants* that isolate uncertainty sources while keeping game logic fixed, e.g.: (i) sticky-action on/off (or multiple $p$ values), (ii) deterministic vs. stochastic reset regimes, and (iii) single-agent vs. multi-agent variants when available. Similarly, when a benchmark has both a public Markov interface and a partially observed one (e.g., board state vs. pixels), reporting TCP on both makes the interface dependence explicit (Theorem 1).

**Suite metadata.** A TCP-spanning suite should ship with a canonical $d_{\mathrm{probe}}$ definition (scripts/policies) and published TCP-Std-v1 numbers for each benchmark-protocol-interface triple, so model papers can cite stable reference values.

### B.3. Benchmark instrumentation

**Recommendation (make TCP measurable by default).** Benchmarks should expose the minimal hooks needed to compute TCP under the protocols of Sec. 6. This is not "extra analysis"; it is basic experimental control. In particular, instrumentation should support (a) separating uncertainty sources (chance vs. interaction) as motivated by Theorem 2, and (b) reproducing the induced kernel (Sec. 6.4).

**Simulator-backed benchmarks (preferred when feasible).** When the benchmark is a simulator or executable environment, provide:

- **Reset/replay.** Ability to reset to a saved latent state consistent with the declared interface, and replay transitions under the declared protocol.
- **Controllable resampling.** Ability to resample exogenous randomness $\Omega_t$ independently at fixed $(x, a)$ (and, in multi-agent settings, to sample opponents from a declared $\Pi$), enabling unbiased collision-based Axis I estimates (Sec. 6.1).
- **Logged randomness and actions (when available).** If $\Omega_t$ is accessible (tile spawns, shuffles, procedural draws), log it; likewise log opponent actions $B_t$ and, when meaningful, opponent identifiers/types $\theta$. This makes source entropies and influence diagnostics directly computable and interpretable (Theorem 2; Sec. 4.2).
- **Protocol transparency.** If wrappers modify action execution or add stochasticity, expose and log the wrapper

parameters and the executed action sequence; otherwise TCP values cannot be meaningfully compared across implementations.

**Log-only benchmarks (datasets and learned "neural environments").** When only recorded trajectories are available, TCP remains computable via the probe-based proxies of Sec. 6.2, but only if the dataset includes:

- **Complete aligned histories.** The observation stream and action stream needed to construct the declared $X_t = \phi_k(H_t)$ without ambiguity (including preprocessing/tokenization details).
- **Axis-II fields when claimed.** If interaction-induced branching is to be reported, the dataset must log $B_t$ (and ideally $\theta$); otherwise Axis II must be marked unobservable rather than inferred.
- **A fixed evaluation split and scripts.** A canonical held-out set and reference scripts for TCP-Ref probes, so that $\mathcal{L}$ and $\mathcal{L}_k$ are reproducible across papers.

**Outcome.** With these hooks, TCP becomes comparable benchmark metadata (like observation resolution or action-repeat), rather than an analysis that each paper re-derives differently. This is the core community recommendation of the paper: make transition complexity *measured and versioned* so that progress in GWMs is interpretable across environments, protocols, and interfaces.

