# OpenReview forum: "Position: Profiling Game Worlds by Transition Complexity"
_ICML.cc/2026/Position_Paper_Track — ICML 2026 Position Paper Track regular_

### Official Review · Reviewer_6Q1K · 2026-02-25

**Significance:** 4
**Argument Clarity:** 3
**Rating:** 5
**Confidence:** 4

**Questions:**

You mention numbers for the amounts of samples to estimate TCP metrics. Shouldn't the amount of samples be dependent on the complextiy of the transition distribution?

What is the impact of the exploratory policy being used to collect the samples for measuring the TCP score?
How much do the TCP scores depend on this policy and might using strongly varying policies lead to strongly varying TCP scores for practically the same environment?

**Alternative Views Section:**

Yes

**Compliance With Llm Reviewing Policy A Conservative:**

Affirmed.

**Discussion Potential:**

3

**Final Justification:**

The paper touches an important issue about autonomous decision making; How difficult is it to understand an environment?
In general, the paper is good to read and formalizes different views in which an environment can be challenging.

**Paper Summary:**

The paper propose transition complexity profiles for analyzing games and virtual environments. In particular, the authors argue that current metrics do not offer a clear indication how difficult it is to learn a transition model aka world model.
Thus, they propose to compute statistics on three levels. The first axis is about the branching of the transition function which is evaluated by entropy measures on the distribution of one step transitions. The second axis is on branching due to antagonistic agents in the same environment . The third axis is on temporal and spatial dependencies. As in partially observable environments, observation uncertainty might also yield an important challenge for learning a world model, the proposed TCP metrics evaluated the entropies over various histories. Similar ideas can be applied to spatial neighborhoods if applicable for the game at hand.
After defining the metrics the auhors comment on how they might be computed and provide theorems on refining the environments.

**Position:**

Yes

**Position In Title:**

Yes

**Related Work:**

4

**Strengths And Weaknesses:**

Strong points:
* The paper presents a well-founded and broad view of the oririgins of uncertainty in an environment and provide measures on how to evaluate them.
* Though concrete examples are only provided in the appendix, the authors also try to provide recipies for how to compute the proposed TCP benchmarks.

Weak points:
* Though the paper makes clear that existing metrics do not really capture the complexity of the model indepently from a particular goal or policy, the authors could lead a more convincing argumentation about what the advantages of knowing the TCP metrics for a given environment and or offline RL dataset are.
* The paper reads at some point rather technical and could use a running example to better follow the definitions and terminology.

**Support:**

3

---

> ### Author Rebuttal · Authors · 2026-03-27
>
> Thank you for the supportive and constructive review. We are encouraged that you found the paper broad, well-founded, and practically oriented, and we appreciate the suggestion that the argument for why TCP matters could be made even more concrete.
>
> **1) "Why is it useful to know TCP for an environment or offline dataset?"**
> Our core claim is that current world-model evaluation often reports *how well a model did* without reporting *what transition problem it was asked to solve*. TCP is intended to supply that missing denominator. In practice this matters in at least three ways:
> - it makes cross-benchmark claims interpretable (e.g., two environments may have similar return scores but very different branching/span profiles);
> - it makes protocol/interface changes visible (e.g., sticky actions, tokenization, context windows change the induced kernel); and
> - it helps benchmark design by ensuring suites span different kinds of transition difficulty rather than only different modalities.
>
> We agree that a running example helps communicate this, and in the revision we will move the intuition from the tic-tac-toe worked example forward so the definitions feel less abstract.
>
> **2) "Should sample counts depend on complexity of the transition distribution?"**
> Yes in a statistical sense, but the fixed TCP-Lite-v1 / TCP-Std-v1 counts are meant as **versioned comparable budgets**, not as a claim that every environment is equally easy to estimate. In harder/high-branching regimes, the correct outcome may be that the estimate is **collision-limited at that tier** (as the paper already specifies), together with bootstrap confidence intervals and distributional summaries (mean/median/90th percentile). In other words, the fixed tiers provide comparability; they do not pretend to guarantee the same estimator variance for every benchmark. We will make this point more explicit.
>
> **3) "How much do TCP scores depend on the exploratory policy?"**
> Potentially a lot, which is precisely why the paper requires TCP to be reported under **both** $d_{\mathrm{data}}$ and a standardized $d_{\mathrm{probe}}$. Different policies can visit very different transition regimes of the same environment. We view that as signal, not noise: it is often scientifically important to know whether a model is being evaluated on a narrow easy slice versus a broad/high-branching slice of the kernel. Standardization comes from matching the interface, protocol, and $d_{\mathrm{probe}}$, while $d_{\mathrm{data}}$ reveals what the model actually saw during training. We will sharpen this dual-reporting rationale in the paper.
>
> Thank you again for the thoughtful comments. They help us improve the paper's explanatory clarity without changing its main position: transition complexity should be measured and reported explicitly, rather than left implicit behind benchmark names alone.

---

### Official Review · Reviewer_Sv24 · 2026-03-12

**Significance:** 2
**Argument Clarity:** 3
**Rating:** 4
**Confidence:** 3

**Questions:**

1.  Training the reference probes for 200M to 1B tokens across multiple seeds represents a significant compute investment for researchers simply trying to report benchmark metadata. How do you envision mitigating this cost to encourage widespread adoption, particularly for smaller academic labs?

2. Do you plan to release a centralized repository or empirical study detailing the computed TCP-Std-v1 values for standard environments (e.g., the ALE suite, popular DeepMind Control Suite tasks)?

3.  You note that multi-actor effects often appear as latent uncertainty when other controls are not logged, and dictate that Axis II should be marked unobservable in these cases. What is your recommendation for researchers working with raw video datasets where Axis II is completely inaccessible.

**Alternative Views Section:**

Yes

**Compliance With Llm Reviewing Policy A Conservative:**

Affirmed.

**Discussion Potential:**

2

**Paper Summary:**

The paper identify a gap in the evaluation of Game World Modeling (GWM) and Reinforcement Learning (RL): while models are frequently evaluated on downstream return or visual generation quality, the intrinsic difficulty of the underlying transition prediction problem is rarely quantified. The author propose the Transition Complexity Profile (TCP), a standardized framework designed to characterize an environment's transition kernel at a declared interface. TCP decomposes transition complexity into three measurable axes. Axis I: Intrinsic one-step branching, measured via collision entropy or predictive coding. Axis II: Interaction-induced branching, which quantifies the uncertainty stemming from opponet policies, population heterogeneity, and their actual influence on the next state. Axis III: Temporal and spatial dependency span, measured by identifying the point at which predictive performance saturates using standardized probe models. The paper advocates the position that TCP should become standard benchmark metadata in GWM and RL research. The authors argue that doing so will enable fair, interpretable comparisons across different domains.

**Position:**

Yes

**Position In Title:**

No

**Related Work:**

3

**Strengths And Weaknesses:**

Strengths:

1.  As the ICML community sees a surge in interactive generative simulators and world models, the lack of standardized complexity metrics is a bottleneck. The position advocated by the authors addresses a timely scientific question.

2. Unlike purely theoretical complexity measures, TCP is designed to be measurable. The paper provides concrete instrumentation instructions, including specified probe architectures and precise measurement budgets.

3. The authors identify that transition stochasticity is heavily dependent on the chosen interface (e.g., pixels vs. RAM) and protocol (e.g., sticky actions). Theorem 1 formally proves how interface refinement impacts entropy, making a strong case for why the interface must always be declared.


Weaknesses:

1.  While the paper aims for a "low-friction community norm", the required measurement budgets are non-trivial. This computational burden may deter researchers from adopting TCP as routine metadata.

2. The paper explicitly notes that Axis II should be marked "unobservable" if opponent actions are not logged at the declared interface. Given that many popular offline datasets and human gameplay logs do not expose other agents' controls, this critical axis may remain unmeasurable for a lot of existing benchmarks.

**Support:**

3

---

> ### Author Rebuttal · Authors · 2026-03-27
>
> Thank you for the positive and careful review. We are encouraged that you found the question timely, the framework measurable rather than purely theoretical, and the instrumentation details concrete. We also appreciate that you highlighted what we view as a central point of the paper: transition complexity must be declared together with the **interface and protocol**.
>
> **1) "The measurement budgets may still be too expensive for routine use"**
> We agree this is the main practical adoption challenge, and the paper is designed around mitigating exactly that. First, we define two tiers: **TCP-Lite-v1** for low-friction reporting and **TCP-Std-v1** for canonical benchmark publication. Second, the intended division of labor is that **benchmark maintainers publish TCP-Std-v1 once**, under a canonical $d_{\mathrm{probe}}$, and model papers may simply cite those values unless they change the induced kernel (e.g., different tokenization, context window, sticky-action setting, wrapper, etc.). Third, the reference probes are deliberately small and fixed because they are **measurement instruments**, not new architectures to optimize. We will make this deployment model more explicit so the paper does not read as if every paper must retrain 1B-token probes from scratch for every benchmark.
>
> For smaller labs, our intended norm is: cite the benchmark's published TCP-Std-v1 when available; compute TCP-Lite-v1 only for the paper’s own modified interface/protocol or dataset distribution. That is a much lighter burden than universal re-computation.
>
> **2) "Will you release a centralized repository / empirical study for standard environments?"**
> We agree that this would materially improve adoption. The natural rollout is a central benchmark-card repository collecting canonical TCP values (at least under a fixed public interface and $d_{\mathrm{probe}}$) together with reference code for the TCP-Ref estimators and probes. We will make this recommendation explicit in the adoption section, because community uptake depends on shared infrastructure, not just definitions.
>
> **3) "What should researchers do when Axis II is inaccessible in raw video datasets?"**
> Our recommendation is to be explicit and conservative: **mark Axis II as unobservable** rather than inferring opponent actions from pixels. We view this as a feature, not a weakness, because it prevents brittle or non-reproducible decomposition of uncertainty sources. In such datasets, Axes I and III remain informative: latent multi-actor effects still appear as effective next-state branching at the declared interface, and finite-history requirements are still measurable via the standardized probe curves. We will make the "raw video / no $B_t$ => Axis II unobservable" guidance even more explicit in the final version.
>
> More generally, one purpose of TCP is to distinguish between *not difficult* and *not instrumented*. If a benchmark does not expose the variables needed to separate interaction-induced branching, that should itself be visible in benchmark metadata rather than hidden behind a single aggregate result.
>
> Thank you again for the constructive review. Your comments help us sharpen the paper's practical adoption story.

---

> > ### Author Rebuttal · Reviewer_Sv24 · 2026-04-02
> >
> > Thanks for the response and clarification. I maintain my original positive rating of 4.

---

### Official Review · Reviewer_FVjk · 2026-03-13

**Significance:** 3
**Argument Clarity:** 3
**Rating:** 4
**Confidence:** 3

**Questions:**

Can you provide some empirical results to demonstrate the usefulness of TCP?

Do you plan to release the code?

It would also be nice to have a repository to gather the reported TCP results.

**Alternative Views Section:**

Yes

**Compliance With Llm Reviewing Policy A Conservative:**

Affirmed.

**Discussion Potential:**

3

**Final Justification:**

I recommend accepting this position paper. Although the justification of the proposed transition complexity profile (TCP) is limited, the authors' rebuttal was satisfactory. This is a timely topic and has the potential to open discussion.

**Paper Summary:**

This paper focuses on the prediction of the transition model in the model-based reinforcement learning of games. This paper pointed out the issue in the evaluation protocol in recent model-based RL approaches. That is, the performance is measured solely by the performance on the downstream tasks. It does not reveal how complex the transition prediction task inside the MbRL is. This paper argues that one should report a standardized transition complexity profile for each game environment. This paper proposes complementary metrics to capture the difficulties of the transition prediction task from several perspectives.

**Position:**

Yes

**Position In Title:**

Yes

**Related Work:**

2

**Strengths And Weaknesses:**

Strength

- The topic is timely and important in the recent MbRL trend.

Weakness

- The justification of the proposed transition complexity profile (TCP) is discussed, but is not demonstrated empirically. Although this is a position paper, if a new metric is proposed, it should be justified.

- As a paper proposing a novel evaluation protocol, I think it is essential to release the code. However, it hasn't been done.

**Support:**

2

---

> ### Author Rebuttal · Authors · 2026-03-27
>
> Thank you for the thoughtful review and for recognizing the timeliness of the topic. We agree with the spirit of your comment: if TCP is to be useful as a community standard, it must be concrete, reproducible, and eventually easy to run.
>
> **1) "The metric is proposed but not empirically justified."**
> This is a fair aspect. Our contribution in this position paper is to argue for a **measurement layer** and to make it precise enough that the community can actually adopt it. For that reason, the paper is not purely conceptual: it already contains (i) a fully worked numeric tic-tac-toe TCP example, (ii) a closed-form sticky-action sanity check showing protocol sensitivity, (iii) fixed reference probes / budgets, and (iv) a minimal reporting table and instrumentation recipe. We agree, however, that these concrete pieces should be more prominent. In the revision we will surface the worked example and reporting recipe earlier so the paper reads less as "a new metric name" and more as a reproducible protocol.
>
> More broadly, our claim is not that TCP is the only possible complexity notion, but that papers currently omit benchmark metadata that is essential for interpreting world-model results. Even a lightweight, operational profile of branching / interaction / dependency span would make cross-benchmark claims much more interpretable than today’s practice.
>
> **2) "As a new evaluation protocol, code release is essential"**
> We agree. Because this is a position paper, the most useful artifact is not executable code (yet) but a **reference implementation of the measurement protocol**. Standardization is much more likely if the community has reference estimators rather than re-deriving them from prose. Our intention is to provide a lightweight package containing: collision-based Axis-I estimators, the TCP-Ref probe configs, and a template benchmark-report sheet. If anonymization / venue policy constrains timing, we will still make the release plan explicit in the final version, because we agree this is important for adoption on top of the adoption of our position.
>
> **3) "A repository to gather reported TCP results would be nice"**
> We agree with this as well. In fact, we think the most useful rollout is: benchmark maintainers publish canonical TCP-Std-v1 values under a fixed public interface/protocol/$d_{\mathrm{probe}}$, while individual model papers cite those values and only recompute TCP-Lite-v1 when they change the induced kernel (e.g., tokenization, context window, sticky-action setting, etc.). We will make this "benchmark-card / central repository" idea more explicit in the adoption section.
>
> Because this is a position paper, our primary contribution, so far, is the **standard itself**: a crisp object being measured, a reproducible protocol, and concrete recommendations for how the community could adopt it. We appreciate the push to make that even more actionable, and we will revise the paper accordingly.

---

> > ### Author Rebuttal · Reviewer_FVjk · 2026-04-02
> >
> > The authors have successfully addressed the comments in the initial review. If the promised revisions---specifically the prominence of the worked examples and the explicit plan for the reference implementation---are executed as described, I believe this paper will be a timely and significant contribution to the field of Model-Based RL.

---

### Official Review · Reviewer_rXq7 · 2026-03-15

**Significance:** 3
**Argument Clarity:** 3
**Rating:** 4
**Confidence:** 3

**Questions:**

* Since most of the games use POMDP, where a hidden structure (that could be simple or complex) is always present, would like to know how TCP would be helpful for POMDP, and would it be good to explicitly consider POMDP and show the usage of TCP?

* It would be good if the authors could explain some cases where complexity arises from combinatorial interactions, not just transitions. I am not sure how that part is captured in the proposed metric.

* When TCP is computed on logs, results depend heavily on data coverage. Would that not lead to some dataset bias or environment coverage bias?

* I was also thinking that complexity may also be related to rules generating transitions (mostly considering the definitions of reasoning these days), rather than the distribution of transitions. Considering cases where a simple rule may generate extremely complex behavior (connecting ideas from Kolmogorov Complexity), and I think the Entropy-based metrics cannot capture that completely. It would be great if the authors could explain some limitations on these lines.

**Alternative Views Section:**

Yes

**Compliance With Llm Reviewing Policy A Conservative:**

Affirmed.

**Discussion Potential:**

3

**Final Justification:**

The rebuttal clarifies the proposed TCP’s scope, along with limitations, and interface/policy dependence, addressing some of the initial concerns. These follow-ups regarding interpretation across POMDPs or combinatorial environments, dataset/log coverage bias, and guidance on interface selection for comparability. These points do not undermine the work but would benefit from further clarification. I also liked that the authors clearly acknowledged all the concerns and provided a detailed explanation for the raised concerns. I guess adding these will help improve the quality and impact of this work.

**Paper Summary:**

The paper lays down its position as a recommendation for the RL and Game world-based research community and presents the requirement for reporting/profiling the environment complexity with transition complexity. In general, the paper compares different environments and tries to understand how complex they are, and how this complexity can be quantified to come up with a profiling measure, so that a proper comparison can be made. Primarily, the paper argues that the research community doesn’t have a standardized way to quantify why a world model might perform well on one environment and worse/poor on others, and quantification/standardization is necessary. Further, as a solution, the paper proposes a transition complexity, which uses 3 different criteria to define the complexity, 1) Branching (how many possible futures 2) Interaction (how much other agents matter) 3) Dependency (how far effects propagate).

**Position:**

Yes

**Position In Title:**

Yes

**Related Work:**

3

**Strengths And Weaknesses:**

Strengths

* One of the major strengths of the paper comes from the generic view/formulation of a measure that could be used across domains. TCP resembles past metric revolutions because it standardizes measurement, enables cross-paper comparison, and exposes hidden difficulty in benchmarks. If the usage of this method is accepted by the community, it has the potential to become one way of quantifying complexity, just like perplexity in language models or FID scores in generative modelling.

Weaknesses

* [minor comment] The "Call to Action" section is missing from the paper. As per instructions “We encourage the inclusion of a "Call to Action" section that identifies plausible steps to realizing the aims of the stated position.  By the end of the paper, the reader should know what steps should be taken by whom to bring about these desired outcomes.”It would be good to add these to the paper.

* In general, though the paper explains the position well and present a nice framework, some clear guidelines (or demonstrations) are still lacking, it would be good if the authors could lay out clear guidelines/actions so that the community starts using this metric, making the adoption of this metric easy.

* Looking at the proxies presented in the paper, it seems the proposed metric, TCP, somewhat measures uncertainty, and not functional complexity. This distinction becomes a little important fundamentally, more specifically, if we look at the definitions coming from Information Theory and Computational Complexity Theory. And Entropy may not directly equate to algorithmic complexity. A simple example that comes to mind is an environment that captures something like deterministic but chaotic physics, or a low entropy but extremely nonlinear aspect. In such cases, the TCP may not be a good measure, and a model might learn a stochastic but simple distribution easier than a deterministic but complex mapping. It would be good if the authors could also explain such cases where the proposed position fails.

* Another thing that is a little unclear and leaves room for discussion is that the formulation presented is highly dependent on the used representations, i.e., the proposed formulation defines complexity relative to the interface, which may mean that two different works using different measures for the same environment. For example, a game like chess, the board state is deterministic, and if we solve it in another way, pixel observation is stochastic.  So TCP partly measures perceptual representation difficulty, not environment dynamics, and has a risk of going into domains where TCP describes the learning problem, not the world itself.

* Another thing that may come up in using TCP is its dependence on the state-action distribution used for measurement, making it conditional on the used policy, which is a little unclear, since it now also measures behavior distribution, not inherent environment difficulty, making the comparison again difficult. In general, Complexity becomes policy-relative, which should not be the case, given the motivation for standardization.

* Considering the quantification in TCP is environment-centric, it somewhat also ignores/underestimate strategic complexity. In some of the games, some initial play affects the gameplay over a long horizon. The authors should consider adding some aspect of the long-horizon problem in RL with the proposed TCP pipeline.

* When using the entropy in the formulation, it comes with a risk of missing the underlying semantic structure. Entropy treats all outcomes equally, but in some games, some outcomes are semantically similar (Two states might differ by a few pixels but represent the same game situation, and Entropy counts them as distinct outcomes).

**Support:**

2

---

> ### Author Rebuttal · Authors · 2026-03-27
>
> Thank you for the careful and constructive review. We are encouraged that you see TCP as a generic, cross-domain measurement layer with the potential to play a role analogous to perplexity/FID for transition modeling. The key clarification is that TCP is **not** intended as a universal notion of "game complexity" or algorithmic/Kolmogorov complexity. It is an **operational profile of the induced next-state prediction problem** at a declared interface.
>
> **1) "A Call to Action / clearer guidance is missing"**
> We agree this should be more explicit in the main paper. The current draft already contains concrete adoption steps in the "Landscape context and adoption" section and the appendix "Research agenda and community recommendations": a required kernel declaration, a minimal TCP table, TCP-Lite-v1 vs TCP-Std-v1 tiers, a division of labor between benchmark maintainers and model papers, and recommended instrumentation hooks. In the final version we will elevate this into an explicit **Call to Action** subsection in the main paper.
>
> **2) "TCP measures uncertainty, not functional/algorithmic complexity"**
> We agree this distinction is important. Entropy / collision entropy do **not** equal computational complexity, Kolmogorov complexity, or solver difficulty. A deterministic but highly nonlinear/chaotic mapping can be hard for a model while having low one-step entropy. Our claim is more specific: TCP measures **branching, interaction-induced uncertainty, and finite-history/span requirements** of the transition kernel *at the declared interface*. These are exactly the confounds that are currently unreported in GWM/RL benchmarking. We will add an explicit limitation paragraph stating that TCP is **complementary to**, not a replacement for, algorithmic/rule-description complexity, strategic/planning complexity, or architecture-specific approximation difficulty.
>
> **3) "TCP is representation-dependent; it may describe the learning problem, not the world itself"**
> Yes, and this is intentional. A board-state model and a pixel model are solving different induced kernels, so they should not be assigned the same transition profile. The paper already states that TCP is a property of **(environment, interface, reference distribution)**, and Theorem 1 formalizes the ordering under deterministic coarsening. For comparability, we recommend reporting TCP (i) at the model's native interface and (ii) when available, also at a public community-standard interface. So TCP does not hide representation dependence; it makes it explicit and reproducible.
>
> **4) "TCP depends on the state-action distribution / policy"**
> Also yes, which is why the paper requires reporting **both** $d_{\mathrm{data}}$ and a standardized $d_{\mathrm{probe}}$, together with median/90th-percentile statistics. Different policies can visit different transition regimes of the same environment; collapsing this into a single supposedly policy-invariant number would hide exactly the phenomenon we want to expose. Standardization here means: same interface, same protocol, same declared $d$. We will sharpen this wording to avoid suggesting that TCP is an environment-only constant.
>
> **5) POMDPs, combinatorial interactions, and long-horizon strategic difficulty.**
> POMDPs are actually where TCP is especially useful: hidden variables appear as effective branching in Axis I at finite history, and Axis III quantifies the smallest context length needed for predictive saturation. This is already formalized in the preliminaries via $X_t=\phi(H_t)$ and the operational Markov approximation $P(X_{t+1}\mid X_t^{(k)},A_t)$, but we agree it should be foregrounded more clearly. Likewise, combinatorial interactions matter when they induce larger branching or longer dependency span. What TCP does **not** claim to capture by itself is the full game-theoretic/planning difficulty of acting optimally over long horizons. That is complementary, and should still be reported via return/control metrics.
>
> **6) "Entropy misses semantic equivalence; log-only TCP may reflect dataset coverage"**
> We agree. For log-only settings, TCP is explicitly a property of the **gameplay dataset at the declared interface**, which is why the paper repeatedly says "environment or gameplay dataset" and requires $d_{\mathrm{data}}$ vs $d_{\mathrm{probe}}$. And if semantically equivalent states should be merged, TCP should be computed on a symbolic/tokenized interface that performs that abstraction; if computed on pixels, it intentionally measures the pixel-level prediction problem.
>
> We appreciate these concerns because they help us state the scope more crisply. We will revise the paper to make the limits explicit rather than implicit.

---

> > ### Author Rebuttal · Reviewer_rXq7 · 2026-04-04
> >
> > Thank you for your detailed response. The rebuttal clarifies the proposed TCP’s scope, along with limitations, and interface/policy dependence, addressing some of the initial concerns. I still have follow-ups regarding interpretation across POMDPs or combinatorial environments, dataset/log coverage bias, and guidance on interface selection for comparability. These points do not undermine the work but would benefit from further clarification. I also liked that the authors clearly acknowledged all the concerns and provided a detailed explanation for the raised concerns. I guess adding these will help improve the quality and impact of this work.
> >
> > I will reconsider/raise my score from 3 to 4.

---

### Decision · Program_Chairs · 2026-04-30

**Decision:**

Accept (regular)

**Comment:**

The paper has positive reviews: 3x Borderline Accepts and 1x Accept.
The reviewers generally believe that this is a good and timely paper. There have been some concerns and weaknesses mentioned in the initial reviews, but the reviewers are satisfied after the rebuttal (we have one increase in score and the rest kept their positive scores). Therefore, I recommend its acceptance.

The paper becomes stronger if it
- adds an explicit Call for Action section
- adds some worked examples
- suggests a plan for the reference implementations